# Outcome of Fetal Dysrhythmias with and without Extracardiac Anomalies

**DOI:** 10.3390/diagnostics13030489

**Published:** 2023-01-29

**Authors:** Stephanie Springer, Eva Karner, Elisabeth Seidl-Mlczoch, Guelen Yerlikaya-Schatten, Petra Pateisky, Barbara Ulm

**Affiliations:** 1Department of Obstetrics and Gynecology, Division of Obstetrics and Feto-Maternal Medicine, Medical University of Vienna, Spitalgasse 23, 1090 Vienna, Austria; 2Department of Pediatric and Adolescent Medicine, Division for Pediatric Cardiology, Medical University of Vienna, 1090 Vienna, Austria

**Keywords:** arrhythmias, tachyarrhythmias, bradyarrhythmias, extrasystoles, AV block, extracardiac anomalies, fetal echocardiography, prenatal diagnosis

## Abstract

Fetal dysrhythmias are common abnormalities, which can be categorized into three types: rhythm irregularities, tachyarrhythmias, and bradyarrhythmias. Fetal arrhythmias, especially in high-risk pregnancies, require special monitoring and treatment. The aim of this study was to assess the stillbirth and early and late neonatal mortality rates for pregnancies complicated by fetal dysrhythmias from one single tertiary referral center from 2000 to 2022. Of the 1018 fetuses with congenital heart disease, 157 (15.42%) were evaluated in this analysis. Seventy-four (46.7%) fetuses had bradyarrhythmias, 51 (32.5%) tachyarrhythmias, and 32 (20.4%) had rhythm irregularities. Additional structural heart defects were detected in 40 (25.3%) fetuses and extracardiac anomalies in 29 (18.4%) fetuses. Thirteen (8.2%) families opted for termination of the pregnancy. Eleven (7.6%), out of 144 continued pregnancies ended in spontaneous intrauterine fetal death (IUFD). Neonatal death was observed in nine cases (5.7%), whereas three (1.9%) died within the first 7 days of life. Although most intrauterine fetal deaths occurred in pregnancies with fetal bradyarrhythmia, neonatal death was observed more often in fetuses with tachyarrhythmia (8.5%). The presence of extracardiac anomalies, congenital heart disease (CHD), and Ro-antibodies are predictive factors for the occurrence of IUFD. Rhythm irregularities without any other risk factor do not present higher risks of adverse perinatal outcome.

## 1. Introduction

Fetal dysrhythmias are common abnormalities that can be observed in 2% of unselected and up to 16.6% of high-risk pregnancies [1,2,3]. Dysrhythmias are abnormalities of the fetal heart rate (FHR) or the rhythm (regular or irregular) [4]. Ischemia, inflammation, disturbances of the electrolyte balance, structural heart disease, and gene mutations are held responsible for their occurrence [5]. Clinically, fetal dysrhythmias can be classified into three types: rhythm irregularities, tachyarrhythmias, and bradyarrhythmias [6]. Benign arrhythmias, mainly caused by isolated atrial ectopics, are relatively common, usually resolve spontaneously and therefore usually do not depend on any therapy before or after birth, whereas life-threatening fetal arrhythmias are relatively rare (incidence 1:5000 pregnancies) [7]. The latter, which include supraventricular tachycardias (SVTs, ~3/4 of cases), atrial flutter (AF), ventricular, junctional, or chaotic atrial tachycardias, and bradyarrhythmias, can cause cardiac dysfunction, heart failure, fetal hydrops, and lead to fetal death [7,8,9]. Dysrhythmias associated with persistence and hemodynamic compromise (e.g., cardiomegaly, hydrops, AV-valve regurgitation) require treatment, as they can cause preterm delivery [10]. Therefore, prenatal treatment is reasonable to improve the fetal survival rate. The importance of treating fetal dysrhythmias has progressed over the past three decades [11,12]. In utero, a detailed analysis about the type of arrhythmia is possible using M-mode and Doppler-echocardiography [11]. Tachycardia is defined as FHR > 180 beats per minute (bpm), but persistent heart rates between 160 and 180 bpm may also be classified as abnormal [4]. Depending on the mechanism, hemodynamic impact, fetal well-being, the gestational age, and the parents’ choice, the management options are no intervention, drug therapy, or delivery [4]. In case of tachyarrhythmias, the efficacy of the maternal transplacental administration of antiarrhythmic agents such as digoxin, flecainide, sotalol, and amiodarone has been demonstrated [8,9,13]. If maternal transplacental treatment fails, the direct application of antiarrhythmic drugs such as intraumbilical, intraperitoneal, or intramuscular injection may be considered as an alternative [8,10]. Survival rates are 80–90%, whereby hydrops are the most important determinant of outcome [8,9,14,15]. Fetal bradycardia, mainly due to AV block, is defined as fetal ventricular heart rate lower than 100 bpm. Approximately 50% of all cases originate from associated congenital heart disease (CHD), where prognosis is described as generally poor with high perinatal mortality [11,16,17]. The remaining cases with normal cardiac structure are often caused by maternal SS-A antibodies [11]. The outcome of isolated AV block seems better [17,18,19,20]. Compared to the treatment of fetal tachycardia, the efficacy of prenatal treatment for fetal AV block is limited. Beta-sympathomimetic agents, intravenous gamma globulin (IVIG), plasmapheresis, and steroids have been proven to be effective transplacental treatments to reduce or prevent myocardial and conduction abnormalities, reduce the levels of maternal antibodies, or augment FHR, although established autoimmune AV block III is irreversible [11,21,22,23].

The aim of this study was to assess the stillbirth and early and late neonatal mortality rates for pregnancies complicated by fetal dysrhythmias from one single tertiary referral center for prenatal medicine over the last two decades.

## 2. Materials and Methods

This single center retrospective cohort study was performed at the Department of Obstetrics and Gynecology, Division of Obstetrics and Feto-maternal Medicine, a tertiary referral center for fetal medicine and CHD at the Medical University of Vienna, Austria. The study was authorized by the institutional ethics committee (1085/2020).

Medical data from all patients who were assigned for fetal echocardiography to our center from 2000 to 2022 and whose fetuses were prenatally diagnosed with dysrhythmias were extracted from our fetal CHD database. Inclusion criterion for this study was the prenatal diagnosis of clinically relevant fetal dysrhythmia. With the exception of pregnancies lost to follow-up, all cardiac and extracardiac diagnoses were based on a combination of pre- and postnatal assessment, with postnatal confirmation either by echocardiography, cardiac surgery, additional imaging, or autopsy (dysrhythmias excluded). 

Maternal and pregnancy characteristics were extracted from the obstetric and fetal CHD databases. Extracardiac anomalies were detected prenatally by targeted ultrasound examinations, following fetal echocardiography according to the ISUOG guidelines [24]. The following sonographic abnormalities were counted as soft markers: nuchal edema, single umbilical artery, hyperechoic or dilated intestine, choroid plexus cyst, absent or hypoplastic nasal bone as well as polyhydramnios and oligohydramnios. Routine prenatal cytogenetic analysis such as karyotyping, fluorescence in situ hybridization (FISH), and over the last years, chromosomal microarray analysis (CMA) and whole exome sequencing (WES), was offered in all cases. In specific cases, analysis for DiGeorge or Noonan syndrome was performed. Clinical records were reviewed for information on pregnancy and neonatal outcome such as gestational age at delivery, birth weight, pre- and perinatal mortality, and mortality until one year after life birth.

Assessment of fetal dysrhythmias was performed using M-mode imaging (M-mode) and pulsed-waved Doppler (PWD) echocardiography according to the recommendations of Carvalho et al. [4]. 

Therapy depended on the fetal condition, arrhythmia characteristics, gestational age, maternal health, and parent’s choice. Delivery with postnatal treatment was considered in the case of fetal decompensation and after 35 weeks of gestation. Before 35 weeks, pharmacological treatment to retrieve sinus rhythm was preferred because the risk associated with premature delivery was believed to outweigh the risks of drug administration. In short, fetuses with bradycardia and maternal Ro antibodies were primarily treated with dexamethasone, while the first line treatment for fetal tachycardia consisted of digoxin, followed by flecainide in some cases. Fetuses with other rhythm irregularities were primarily followed up in weekly or bi-weekly intervals. To monitor maternal and fetal well-being, antiarrhythmic agents were started in the hospital. Further check-ups and adjustment of the medication were performed in an outpatient setting, according to the recommendations of Carvalho et al. [4].

Gestational age was based on first trimester ultrasound screening, whenever available, or on the first dating scan recorded. Stillbirth was considered as fetal death after 19 + 6 gestational weeks (GW), and early neonatal mortality was defined as death within 7 days of live birth [25]. 

Fetal signs of hemodynamic compromise were defined as fetal ascites, pericardial effusion, cardiomegaly, hydrops, and/or AV-valve regurgitation. Late neonatal death was defined as death between 7 days and the end of the first year after live birth. Gestational age at birth was categorized into term (≥37 GW), preterm (28 to <37 GW), and extremely preterm (<28 GW). 

### Statistical Analysis

Statistical analysis was performed with SPSS version 20 for Windows (IBM SPSS Inc., Armonk, New York, NY, USA) and reported as the mean (±standard deviation) for normally distributed continuous variables and median (interquartile range) for non-normally distributed continuous variables. A Kolmogorov–Smirnov test was used to identify non-normally distributed continuous variables. In the case of continuous variables, the two groups were compared using the Student’s *t*-test or Mann–Whitney–U test, as appropriate. Categorical variables were analyzed with the Chi-square test or Fisher’s exact test. To calculate the correlations, Pearson’s correlation was used. The logistic binary regression model was used to test the statistical significance and to evaluate the relationship between different variables. For these multivariate analyses, the regression coefficients Beta, their standard errors, and Wald test are given. Differences were considered statistically significant if the *p*-values ≤ 0.05.

## 3. Results

Out of the 1018 fetuses diagnosed with CHD, 232 (22.8%) matched the inclusion criteria. Twenty-nine twin pregnancies were excluded from the analysis to avoid data falsification due to twin specific complications. Forty-six pregnancies were lost to follow-up. Thus, 157 patients were evaluated in this analysis. Hereby, 74 (47.1%) fetuses had prenatally diagnosed bradyarrhythmias, 51 (32.5%) tachyarrhythmias, and 32 (20.4%) had rhythm irregularities (e.g., extrasystoles) (Table 1).

Fetuses with bradyarrhythmias were most likely to have an additional structural heart defect (33.8%). Hereby, the most common CHDs were an isolated ventricular septal defect (28%) and coarctation of the aorta (24%). Only eight (15.7%) fetuses with tachyarrhythmia had a structural CHD. The most common were the isolated ventricular septal defect (37.5%) and the CHD, which was not further specified (37.5%). In the case of rhythm irregularities, seven (21.9%) fetuses were affected by a structural CHD. The most common CHD in this group was an Ebstein anomaly (42.9%). The distribution of CHDs according to the type of dysrhythmia is given in Table 2.

Fetuses with bradyarrhythmia (28.4%) were most likely to show signs of hemodynamic compromise, compared to fetuses with tachyarrhythmia (19.2%) and rhythm irregularities (15.6%). The distribution of fetal signs of hemodynamic compromise according to the type of dysrhythmia is given in Table 3.

Extracardiac anomalies (ECA) were diagnosed prenatally in 29 (18.5%) pregnancies. Fetuses with bradyarrhythmia had the highest rates of ECA (31.1%). Only six (11.8%) fetuses with tachyarrhythmia showed extracardiac anomaly and no fetus with rhythm irregularities was affected. 

Additionally, in 38 (24.2%) cases, sonographic soft markers were found. A detailed distribution of ECAs and soft markers is presented in Table 4. Intrauterine growth restriction (IUGR) at the time of CHD diagnosis was present in 11 fetuses with bradyarrhythmia (14.9%), four fetuses (7.8%) with tachyarrhythmia, and in three (9.4%) fetuses with rhythm irregularities. 

Fetal magnetic resonance imaging (MRI) was offered in patients with suspicion of additional extracardiac abnormalities. Mainly patients with fetal bradyarrhythmia (27%) received an MRI scan, as the frequency of extracardiac malformations was increased in the bradyarrhythmia subgroup. 

Genetic testing was performed in 23 (31.1%) pregnancies with fetal bradyarrhythmia (amniocentesis: 10, chorionic villus sampling: 10, postpartum sampling: 2, non-invasive prenatal testing: 2), 12 (23.5%) pregnancies with tachyarrhythmia (amniocentesis: 6, chorionic villus sampling: 5, postpartum sampling: 1), and two (6.3%) pregnancies with rhythm irregularities (amniocentesis: 1, chorionic villus sampling: 1). In some patients, multiple procedures were used to obtain genetic material. A total of seven (4.5%) fetuses were genetically abnormal (bradyarrhythmia: 4 fetuses, tachyarrhythmia: 2 fetuses, rhythm irregularities: 1 fetus).

In total, 13/157 (8.3%) families opted for the termination of pregnancy (TOP). TOP was performed in 12.2% (9/74) of pregnancies with bradyarrhythmia, in 5.9% (3/51) of pregnancies with tachyarrhythmia, and in 3.1% (1/32) of pregnancies with rhythm irregularities (Table 1). Univariate analysis showed that the body mass index (BMI) (Pearson correlation −0.169, *p* = 0.05), the gestational age of CHD diagnosis (Pearson correlation −0.386, *p* < 0.01), the presence of CHD (Pearson correlation 0.191, *p* = 0.02) or an extracardiac anomaly (Pearson correlation 0.331, *p* < 0.01), and an abnormal genetic testing result (Pearson correlation 0.544, *p* < 0.01) correlated with the decision of TOP. Multivariate analysis demonstrated that only the gestational age of diagnosis of CHD (Beta = −0.036, S.E. = 0.015, Wald = 6.17, *p* = 0.013), an abnormal genetic testing result (Beta = 6.26, S.E. = 2.26, Wald = 7.66, *p* = 0.006) and the maternal BMI (Beta = −0.33, S.E. = 0.16, Wald = 3.91, *p* = 0.05) significantly influenced the decision of TOP.

In 144/157 (91.7%) pregnancies, the parents had the intention to continue the pregnancy. Eleven (7%) pregnancies ended in spontaneous intrauterine fetal death (IUFD). IUFD/death during delivery occurred in 10 (13.5%) fetuses with bradyarrhythmia, in one (2%) with tachyarrhythmia, and none in fetuses with rhythm irregularities (Table 1). The presence of ECAs (Pearson correlation 0.253, *p* = 0.02), especially anomalies of the nervous system (Pearson correlation 0.182 *p* = 0.02), anomalies of the digestive system (Pearson correlation 0.292, *p* < 0.001) and anomalies of the limbs (Pearson correlation 0.508, *p* < 0.001) correlated with the occurrence of IUFD. Additionally, the presence of a CHD (Pearson correlation 0.178, *p* = 0.03), the presence of bradyarrhythmia (Pearson correlation 0.244, *p* = 0.002) and Ro-AK (Pearson correlation 0.467, *p* = 0.03) correlated with the occurrence of IUFD. No correlation was found between the occurrence of IUFD and maternal BMI, maternal age, nulliparity, maternal pre-existing conditions, PROM, IUGR at diagnosis, signs of cardiac decompensation, and fetal sex.

Maternal and gestational complications are listed in Table 5. 

Maternal immunologic disorders were most prevalent in the bradycardia group (14.9%). Maternal antinuclear antibodies against the Ro-antigen (SSA) were found in seven patients (9.4%). Six patients (8.1%) received dexamethasone. Patients who received dexamethasone had a significantly lower gestational age at birth (Pearson correlation −0.361, *p* = 0.008) with a median gestational age of 34.7 weeks (32–35.3) compared to the gestational age of 38 weeks (35–39.4) in patients without dexamethasone treatment. We also observed lower birthweight (Pearson correlation −0.307, *p* = 0.03) with a median birthweight of 1873 g (1630–1690) with dexamethasone therapy, whereas the median birthweight in patients without transplacental therapy had a median birthweight of 2968 g (2013–3287). The treatment with dexamethasone did not have an influence on neonatal deaths or IUFD.

A total of 29.4% fetuses with tachyarrhythmia were treated with transplacental antiarrhythmic medication. All 15 received digoxin either intravenous or orally, while three (5.9%) patients were additionally treated with flecainide. The transplacental treatment with digoxin did not show any correlation with neonatal survival, gestational age at birth, Apgar, umbilical cord pH, or birthweight. One patient (3.1%) with fetal rhythm irregularities received digoxin and flecainide.

Overall, 133/ 144 (92.4%) of pregnancies, after excluding TOPs, ended in livebirths. A total of 48.1% of live born children were male and the remaining 51.9% were female. A total of 38 children were born preterm before 37 weeks of gestation, mainly iatrogenic due to fetal deterioration. Neonatal deaths were observed in nine cases (6.3%). The highest neonatal mortality rate was observed in the cohort of tachyarrhythmias (8.5%), followed by rhythm irregularities (6.5%) and bradyarrhythmias (5.6%). More neonates died after the first week of life (6/9) than in the early neonatal period. While the cohort of bradyarrhythmias had the lowest neonatal death rate, more neonates from this group were transferred to the neonatal intensive care unit after birth (61.8%), had lower Apgar scores, and were more often diagnosed with arrhythmias postpartum (72.7%). In general, children assigned to this cohort were born at earlier weeks of gestation and had lower birthweights. Further birth characteristics are presented in Table 6. 

Neonatal death correlated with signs of hemodynamic compromise (Pearson correlation 0.188, *p* = 0.025) and lower gestational age (Pearson correlation −0.444; *p* < 0.001). When multivariate analysis was conducted, signs of hemodynamic compromise did not show a significant influence on neonatal deaths, whereas the male gender (Beta = −3.27, S.E. = 1.50, Wald = 4.71, *p* = 0.04), the occurrence of congenital heart disease (Beta = 3.33, S.E. = 1.67, Wald = 4.0, *p* = 0.05), and the gestational age at birth (Beta = −0.055, S.E. = 0.02, Wald = 6.23, *p* = 0.013) had an impact on the neonatal mortality rates. Maternal age, BMI, parity, type of arrhythmia, ECA, IUGR, or the presence of Ro-AK did not show any correlation with neonatal mortality. 

Of the 55 (74.3%) livebirths in the bradyarrhythmia cohort, one neonate died within the first week and two in the late neonatal period. The early neonatal death occurred in a patient with a fetal complete AV block grade III, diagnosed at gestational week 20 + 6. The child also had hypertrophic cardiomyopathy and heterotaxy syndrome. During pregnancy, it developed fetal hydrops and died ten minutes after birth. One case presented prenatally with a tumor in the left heart near the ventricular septum of 7 × 6 × 4 mm and severe bradycardia of 35 bpm during the first fetal echocardiography. Further examinations led to the diagnosis of a hypoplastic left ventricle due to the obstruction caused by the tumor. The parents decided against genetic testing and wanted to continue the pregnancy, despite the poor prognosis. The child was born in gestational week 38 + 6 with an Apgar score of 5/7/7. The newborn was resuscitated after cardiac arrest two days postpartum and required induced therapeutic hypothermia. Two months after birth, the child received a heart angiography with anastomosis surgery and was connected to an extracorporeal heart–lung machine; she also required a pacemaker implantation. Four months after delivery, the infant died due to acute heart failure. Autopsy was declined by the parents. In one patient, premature rupture of the membranes occurred in week 22 + 1. After two weeks of intensive observation, bradycardia (bigeminy) was observed. The child was born with a gestational age of 25 + 6 weeks in the presence of a maternal amnion infection syndrome. The boy developed disseminated intravascular coagulation and subsequently suffered from multiorgan dysfunction syndrome, which led to his demise 22 days after birth.

Forty-seven fetuses with tachyarrhythmia (92.2%) were born alive, two children died in the early neonatal period and two in the late neonatal period. In one patient of our collective, maternal toxoplasmosis infection during pregnancy was found. The mother was treated with sulfadiazine and pyrimethamine. Childbirth occurred in gestational week 28 + 5 with the presentation of an exstrophy of the bladder and hydrocephaly. The neonate died on the second day postpartum due to severe neonatal sepsis, probably caused by prematurity. In the histological analysis of the autopsy, no toxoplasmosis cells were found. Another neonatal death also appeared to be caused by prematurity. A 550 g child was born in gestational week 23 + 4 and died four days after delivery with global respiratory insufficiency and cardiac decompensation.

Two neonatal deaths of 31 live born fetuses occurred in the cohort of rhythm irregularities. Both children had severe malformations associated with Ebstein anomaly and died with cardiac decompensation as a result of the severe congenital heart defect.

The characteristics of the other neonatal deaths are shown in Appendix A.

## 4. Discussion

This retrospective cohort study at a tertiary referral center for the fetal medicine evaluated perinatal outcome of 157 patients with fetal dysrhythmia. Cohorts included rhythm irregularities (32/157), tachyarrhythmia (51/157), and bradyarrhythmia (74/157). 

Thirty-two patients in our collective were diagnosed with fetal rhythm irregularities such as intermittent supraventricular extrasystoles. Intermitted arrhythmias without atrioventricular block or transition to tachyarrhythmia usually do not require prenatal therapy as they resolute spontaneously [4,26]. Benign arrhythmias constitute 96% of fetal dysrhythmias [27]. In our collective, these cases were underrepresented, as predominantly, patients with fetal arrhythmias in need of prenatal treatment or close observation present at our perinatal center. Cardiac decompensation only occurred in fetuses with additional CHD. Two children died after delivery, both were diagnosed with Ebstein anomaly and severe structural heart defects. In general, however, our study showed a good perinatal outcome for this cohort. One patient opted for the termination of pregnancy due to an additional complex CHD, while the remaining pregnancies ended in live births. Only six children were born preterm, otherwise no abnormalities in the perinatal parameters were observed. An association of irregular rhythm and extracardiac malformations was not present, and transition from benign to severe dysrhythmia was never observed. According to other authors, less than 5% of ectopic dysrhythmias can trigger re-entry tachyarrhythmia or can be blocked and transmit to bradycardia [5,7,28]. 

Our results suggest similar outcomes of fetuses with tachyarrhythmia when compared to previously published studies [12,29,30]. Moodley et al. classified tachyarrhythmias in supraventricular tachycardia and atrial fibrillation. They studied 69 pregnancies and reported 94% livebirths of fetuses with atrial fibrillation and 98% livebirths of those with supraventricular tachycardia. A total of 2.9% of the pregnancies ended in intrauterine death and two neonates died postnatally. Postnatal arrhythmias were observed in 63–67% of the children [29]. A study by O’Leary et al. analyzed records of 65 patients with fetal tachyarrhythmia, after excluding cases with fetal CHD. They suggest higher rates of fetal hydrops (20%) and preterm birth (28%). Despite the exclusion of CHD, they describe a neonatal mortality rate of 3% [30]. Fetal tachyarrhythmia can evolve on the basis of various etiologies. Congenital heart defects are described in 1–5% of pregnancies with fetal tachyarrhythmia [4]. In our collective, 15.7% of fetuses with fetal tachyarrhythmia had additional CHD. The findings of systematic reviews reported worse outcome in fetuses with additional structural heart defects [10,26,31]. However, we did not find a significant influence of the presence of CHD on adverse perinatal outcomes in this cohort. Extracardiac malformations, present in 11.8% in our collective of fetuses with tachycardia, are not commonly described in these fetuses. 

The decision for intrauterine fetal treatment depends on several factors such as the type and severity of dysrhythmia, gestational age, associated CHD, and fetal signs of decompensation. Choosing the right therapy can have a major impact on the children’s outcome. In general, the motivation and range for treatment in utero increases with prematurity [32]. In the beginnings of transplacental drug application, digoxin has been established as first-line therapy in the treatment of fetal tachyarrhythmias. For successful digitization, the dose is saturated to a digoxin level of 2–2.5 ng/mL either intravenously or orally, then a maintenance dose of 0.25 mg every 8 h should be applied orally [22,33]. However, digoxin should not be given in cases where fetal hydrops has been diagnosed as the distribution in decompensate fetuses is reduced [4]. Over the years, other medications have shown the effective treatment of fetal tachycardia. Due to higher rates of adverse side effects, some medications are not recommended for standard use. Verapamil has been described with the presence of sudden demise, whereas procainide may trigger preterm delivery. Currently, aside from digoxin, flecanoids and soltalol are used as standard therapy [34]. Flecainide 100 mg every 8 h can lead to the conversion of fetal supraventricular tachycardia [8,9,31,34]. Miyoshi et al. established a protocol for pregnancies without signs of hydrops [12]. Jaeggi et al. initiated the FAST trail, where patients with different types of tachyarrhythmia were randomized and received prespecified therapy protocols [35]. Outcomes will be examined in a prospective analysis [35]. In our collective, 29.4% of fetuses received in utero treatment, whereby all 15 patients were treated with digoxin alone and three additionally with flecainide. Sotalol or amiodarone was not used in our patient collective. We did not find any correlation regarding the transplacental treatment with digoxin and neonatal survival or other favorable perinatal outcomes in the collective of tachyarrhythmias. 

Our results suggest a higher occurrence of IUFD in patients with CHD, which complies with the findings of systematic reviews [4,31]. Structural heart defects can also lead to impaired excitation transmission in the AV node, causing bradycardia. These cases were defined with a fetal heart rate of <100 bpm [11,16]. Bradyarrhythmia represented the majority of fetal arrhythmias in our study with 47.1%. According to our findings, bradyarrhythmia is the only type of dysrhythmia that might lead to a higher rate of IUFDs. 

A total of 33.8% of these were diagnosed with congenital heart defects such as ventricular septum defect, atrioventricular septum defect, transposition of the great arteries or the tetralogy of Fallot. Left isomerism and heterotaxy syndrome are common associations with fetal bradycardia causing AV block [11,36]. In a recently published paper by Seidl-Mlczoch et al., fetuses with heterotaxy syndrome were studied. A total of 24% of those fetuses also presented with bradycardia due to AV block [37]. Associations between isomerism and bradycardia were also be observed in our collective. Most fetuses had heterotaxy syndrome, followed by anomalies of the nervous system, urinary tract, and limb anomalies. In the literature, low survival rates and poor neonatal outcome are described for fetuses with both conditions combined [38]. Our records describe a significant influence of extracardial anomalies on the occurrence of IUFD, especially regarding anomalies of the digestive system, anomalies of the limbs and the nervous system. Escorbar-Diaz et al. investigated perinatal outcomes of those patients, where 79% neonates were born alive, after excluding the families who opted for termination of pregnancy (14%). All patients with AV block required a pacemaker implantation during the neonatal period. The one-year survival rate was reported to be 63% [36]. Lopes et al. presented a similar livebirth rate (75%) but showed a much lower neonatal survival rate (26%) [16]. Our results comply with the findings of Escorbar-Diaz, with 74.3% fetuses born alive and only three neonates who died. However, only 34.6% of these neonates required a pacemaker implantation.

Most patients observed by Lopes et al. were seropositive for antinuclear antibodies (72%). Because more than 90% of infants of seropositive mothers survived the first year of life without prenatal treatment, Lopes et al. criticized the administration of immunosuppressive drugs [16]. This work suggests a significant correlation between the presence of Ro-antibodies and IUFD. Other authors have also described higher rates of morbidity and mortality in patients with fetal arrhythmia in seropositive mothers [21]. A prospective evaluation of fetuses with AV block with and without the administration of dexamethasone was carried out by Friedman et al. in 2009. This study supports the hypothesis of the ineffectiveness of dexamethasone, as the perinatal outcomes in both groups were similar with regard to fetal growth restriction in the DEXA group as a side effect of the treatment [39]. Our findings correspond to these results, as we saw a significant influence of dexamethasone treatment on the birthweight. However, we also observed a lower gestational age at delivery, which in general leads to lower birthweight. Our correlation analysis suggests that transplacental dexamethasone application might lead to delivery at an earlier gestational age. The American Heart Association recommends dexamethasone treatment only in fetuses with severe AV block, while other forms of bradyarrhythmia should be closely observed [40]. One patient received intravenous gamma globulin (IVIG) in addition to dexamethasone without any improvement regarding the perinatal outcome. In general, the use of IVIG to prevent congenital AV block could not be established [39]. 

As we know, structural heart disease can lead to fetal arrhythmia, especially to excitation conduction disorders such as AV block. This work highlights the role of CHD as an unfavorable factor for survival in fetal arrhythmias. Further research with larger case numbers per specific CHD is needed to determine whether the development of arrhythmias in fetuses with CHD leads to higher rates of adverse offspring outcomes compared to fetuses with the same specific CHD but without intrauterine or neonatal arrhythmia. 

## 5. Conclusions

Our findings show that fetuses with rhythm irregularities presented with more favorable perinatal outcomes than bradyarrhythmias and tachyarrhythmias, especially those without CHD. Extracardiac anomalies and CHDs were mainly observed in fetuses with bradyarrhythmia. Neonatal death was the most common in patients with low gestational age at birth. Predictive factors for poor neonatal outcome (higher rates of IUFD) included fetal bradyarrhythmia, the presence of structural heart defects, extracardiac anomalies, and Ro-antibodies. 

## 6. Strengths and Limitations

The retrospective study design may have been source of bias in our work, especially considering the cohort allocation. Our department is certified as aa perinatal center with a focus on feto-maternal medicine. Thus, only selected patients were observed at our department, benign arrhythmias were underrepresented, and patients with uncomplicated clinical course were lost to follow up. Therefore, our collective is not representative for the overall population. Otherwise, rare conditions such as fetal tachyarrhythmias can be detected, observed, and described to gain further knowledge for the diagnostics and the therapy of special patient collectives. This facilitates counseling for the affected patients and leads the way for further research issues. Fetal heart MRI scans have been conducted at our department but have not yet been established as a standard diagnostic method, as fetal echocardiography is still considered the gold standard for the evaluation of cardiac structures and cardiac arrhythmias at our center, at the time when these patients were examined.

## Figures and Tables

**Table 1 diagnostics-13-00489-t001:** Characteristics of the 159 pregnancies with prenatal diagnosis of fetal dysrhythmia.

Variable	Bradyarrhythmias (n = 74)	Tachyarrhythmias (n = 51)	Rhythm Irregularities (n = 32)
Nulliparity ^1^	23 (31.1)	20 (39.2)	14 (43.8)
Maternal age (years) ^2^	31 (5)	30 (6)	30 (6)
Body Mass Index ^2^	26 (5)	27 (5)	27 (5)
Gestational age at diagnosis ^2^	24.4 (7.4)	26.6 (8.6)	30 (6.7)
Structural heart defect ^1^	25 (33.8)	8 (15.7)	7 (21.9)
Termination of pregnancy (TOP) ^1^	9 (12.2)	3 (5.9)	1 (3.1)
Death in utero or during birth ^1^	10 (13.5)	1 (2)	0 (0)
Livebirths ^1^	55 (74.3)	47 (92.2)	31 (96.9)
Beats per minute (bpm) ^3^	75 (58–95)	192 (178–220)	142 (134–160)

^1^ Number (percent), ^2^ Mean (standard deviation), ^3^ Median (interquartile range).

**Table 2 diagnostics-13-00489-t002:** Congenital heart disease (CHD) among fetuses with dysrhythmia.

CHD	Bradyarrhythmias (n = 25)	Tachyarrhythmias (n = 8)	Rhythm Irregularities (n = 7)
Septation defects			
Isolated ventricular septal defect	7 (28)	3 (37.5)	0
Common atrium	2 (8)	1 (12.5)	1 (14.3)
Atrio-ventricular septal defect	5 (20)	0	1
Anomalies of the ventricles			
Hypoplastic left heart syndrome	3 (12)	0	0
Hypoplastic right heart syndrome	1 (4)	0	0
Ventriculo-arterial anomalies			
d-Transposition of the great arteries	2 (8)	0	0
Tetralogy of Fallot	1 (4)	0	0
Double-outlet right ventricle	3 (8)	0	0
Truncus arteriosus communis	1 (4)	0	1 (14.3)
Outflow tract anomalies			
Aortic stenosis/atresia	0	1 (12.5)	0
Coarctation of the aorta	6 (24)	2 (25)	1 (14.3)
Aberrant right subclavian artery	0	0	1 (14.3)
Pulmonary stenosis/atresia	3 (12)	0	1 (14.3)
Complex CHD	4 (16)	1	1 (14.3)
Ebstein anomaly	2 (8)	1 (12.5)	3 (42.9)
CHD not further specified	2 (8)	3 (37.5)	0
Rhabdomyoma	1 (4)	1 (12.5)	0

Data are presented as number (percent). Some fetuses had more than one cardiac anomaly. CHD, congenital heart disease.

**Table 3 diagnostics-13-00489-t003:** Fetal signs of hemodynamic compromise.

Variable	Bradyarrhythmias (n = 74)	Tachyarrhythmias (n = 51)	Rhythm Irregularities (n = 32)
Sign of hemodynamic compromise (total)	21 (28.4)	10 (19.6)	5 (15.6)
Fetal ascites	3 (4.1)	3 (5.9)	1 (3.2)
Pericardial effusion	11 (14.9)	3 (5.9)	1(3.2)
Cardiomegaly	8 (10.8)	2 (3.9)	2 (6.3)
Hydrops	6 (8.1)	3 (5.9)	2 (6.3)

Data are presented as number (percent). Some fetuses had more than one sign of hemodynamic compromise.

**Table 4 diagnostics-13-00489-t004:** Extracardiac anomalies and sonographic soft markers.

Anomaly	Bradyarrhythmias (n = 74)	Tachyarrhythmias (n = 51)	Rhythm Irregularities (n = 32)
Extracardiac anomalies (total)	23 (31.1)	6 (11.8)	0
Nervous system, spina bifida	6 (26.1)	1 (16.7)	0
Digestive system	1 (4.4)	0	0
Urinary tract	3 (13.0)	3 (50)	0
Limb	3 (13.0)	0	0
Heterotaxia	7 (30.4)	1 (16.7)	0
Eye, ear, face	0	1 (16.7)	0
Other	0	1 (16.7)	0
Sonographic soft markers (total)	16 (21.6)	16 (31.4)	6 (18.8)

Data are presented as number (percent). Some fetuses had more than one extracardiac anomaly.

**Table 5 diagnostics-13-00489-t005:** Maternal morbidity: Pre-existing conditions and gestational complications in women with fetal dysrhythmias.

Maternal Morbidity	Bradyarrhythmias (n = 74)	Tachyarrhythmias (n = 51)	Rhythm Irregularities (n = 32)
Pre-existing conditions	21 (28.4)	13 (25.5)	11(34.3)
Cardiac diseases	7 (9.4)	0	3 (9.4)
Immunologic diseases	11 (14.9)	4 (7.8)	1 (3.1)
Antinuclear antibodies	7 (9.4)	0	0
Depression	1 (1.3)	0	0
Diabetes mellitus type I or II	2 (2.7)	2 (3.9)	0
Gestational diabetes mellitus	3 (4.0)	3 (5.9)	4 (12.5)
Pre-existing hypertension	0	1 (2)	0
Hypertensive disorders of pregnancy	0	1 (2)	2 (6.2)
Premature rupture of membranes	12 (16.2)	6 (11.8)	2 (6.2)
Cervical insufficiency	1 (1.3)	1 (2)	1 (3.1)

Data are presented as number (percent).

**Table 6 diagnostics-13-00489-t006:** Birth characteristics.

Variable	Bradyarrhythmias (n = 55)	Tachyarrhythmias (n = 47)	Rhythm Irregularities (n = 31)
Gestational age at delivery—live births (weeks) ^2^	37.9 (34.7–39.3)	38.6 (35.6–40.1)	39.43 (38.0–40.3)
Preterm delivery (≤37.0 weeks) ^1^	19 (34.6)	13 (27.7)	6 (19.4)
Delivery mode caesarean ^1^	36 (65.5)	27 (57.4)	18 (58.1)
Birth weight livebirths (grams) ^2^	3000 (2390–3337)	3310 (2380–3647)	3256 (2970–3640)
Birth weight (percentile) ^2^	32 (17–80)	44 (28–87)	48 (25–75)
Birth length (centimeter) ^2^	48 (42–50)	50 (43–53)	51 (49–53)
APGAR 1 min ^2^	8.0 (7.0–9.0)	9.0 (8.0–9.0)	9.0 (9.0–10.0)
APGAR 5 min ^2^	9.0 (8.0–10.0)	10.0 (9.0–10.0)	10.0 (10.0–10.0)
APGAR 10 min ^2^	9.0 (9.0–10.0)	10.0 (10.0–10.0)	10.0 (10.0–10.0)
APGAR 5 < 7 ^1^	4 (7.3)	3 (6.4)	1 (3.2)
Umbilical cord pH ^2^	7.3 (7.2–7.3)	7.27 (7.17–7.33)	7.30 (7.3–7.4)
Neonatal deaths ^1^	3 (5.6)	4 (8.5)	2 (6.5)
Early neonatal deaths ^1^*	1	1	1
Late neonatal deaths ^1^**	2	3	1
Neonatal survival ^1^	52 (94.6)	43 (91.5)	29 (93.6)
Transfer to neonatology ^1^	34 (61.8)	19 (40.4)	6 (19.4)
Postnatal diagnosis of arrhythmia or congenital heart defect ^1^	40 (72.7)	23 (48.9)	11 (35.5)
Postnatal intervention ^1^	19 (34.6)	7 (14.9)	2 (6.5)
Neurological deficits ^1^	10 (18.2)	4 (8.5)	0

^1^ Number (percent), ^2^ Median (interquartile range); * Early neonatal death = death within one week after birth; ** Late neonatal death = death within 30 days after birth.

## Data Availability

The data presented in this study are available on request from the corresponding author. The data are not publicly available due to the local data protection guidelines.

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
