# Peer review of "Outcome of Fetal Dysrhythmias with and without Extracardiac Anomalies"

_diagnostics, 2023, doi:10.3390/diagnostics13030489_

Round 1
Reviewer 1 Report
Nice work and well written paper, very interesting
Author Response
Thank you very much for acknowledging our work. We are very pleased about your comment. According to your valuable comment, we have now performed both a language and a spell check for the revised version of the manuscript.
Reviewer 2 Report
Dear authors,
Thanks for the paper that is submitted that covers an interesting topic. I, however, have major comments and edits that I would like to suggest to the authors.
After using an anti-plagiarism tool, I have found that there are literally copied paragraphs from this article (in the Methods section):
Springer, S.; Karner, E.; Worda, C.; Grabner, M.M.; Seidl-Mlczoch, E.; Laccone, F.; Neesen, J.; Scharrer, A.; Ulm, B. Outcome after Prenatal Diagnosis of Trisomy 13, 18, and 21 in Fetuses with Congenital Heart Disease. Life 2022, 12, 1223. https://doi.org/10.3390/life12081223
I understand that the Method section can be similar between both articles, but as a suggestion to the authors, I encourage them to change this paragraph in order to avoid the similarities.
The manuscript should undergo a process of proof reading in order to correct some punctuation mistakes, verb correlation…
Line 29: Before introducing CHD, I suggest to use Congenital heart disease.
Line 91: CHD has been previously introduced.
Line 93: What is routine prenatal genetic testing? Caryotype
Line 117: The statistical analysis should be a part of the Methods section.
Line 140: The percentage of bradyarrhythmias was previously introduced with the number 74, in line 135. I think you mean that of the total of bradyarrhythmias, 38.8% had an additional structural heart defect.
Line 164: Sonographic soft markers: Which are them?
Lines 176-182: Why is not the molecular caryotype performed? I understand that in the year 2000 was not an option, but in these past five years is mandatory in a CHD?
Line 218: “in patients with dexamethasone” Redundant.
Line 311: [4][12][29][30][30]. Please check the references
Line 323: [31][10][26]. Please, put them in the proper order.
Line 332: [33][8][9][31]. Please, put them in the proper order.
Line 356: The surname is wrong: Escobar
A conclusion section should be introduced.

Author Response
The manuscript should undergo a process of proof reading in order to correct some punctuation mistakes, verb correlation…
ANSWER: Thank you for this important comment. We made the necessary changes in the Methods section. The manuscript was now proof read by a native speaker.
Line 29: Before introducing CHD, I suggest to use Congenital heart disease.
ANSWER: Thank you for your remark. We first used Congenital heart disease before introducing CHD.
Line 91: CHD has been previously introduced.
ANSWER: Thank you. We changed Congenital heart disease to CHD.
Line 93: What is routine prenatal genetic testing? Caryotype
ANSWER: Thank you for your comment. We changed the sentences.
Routine prenatal cytogenetic analysis like karyotyping, fluorescence in situ hybridiza-tion (FISH), and over the last years chromosomal microarray analysis (CMA) and whole exome sequencing (WES) was offered in all cases. In specific cases, analysis for DiGeorge or Noonan syndrome was performed.
Line 117: The statistical analysis should be a part of the Methods section.
ANSWER: Thank you for this important comment. We agree with you and changed this accordingly. Maybe this mistake happened when the article was inserted into the LaTeX tool.
Line 140: The percentage of bradyarrhythmias was previously introduced with the number 74, in line 135. I think you mean that of the total of bradyarrhythmias, 38.8% had an additional structural heart defect.
ANSWER: Thank you for your comment. In this paragraph we describe the number of additional structural heart defects in fetuses with bradyarrhythmia (25/74 ïƒ 33.8%). For better understanding we changed the sentences.
Fetuses with bradyarrhythmias were most likely to have an additional structural heart defect (33.8%).
Line 164: Sonographic soft markers: Which are them?
ANSWER: Thank you. We inserted following sentence in the Methods section:
The following sonographic abnormalities were counted as soft markers: nuchal edema, single umbilical artery, hyperechoic or dilated intestine, choroid plexus cyst, absent or hypoplastic nasal bone, as well as polyhydramnios and oligohydramnios.
Lines 176-182: Why is not the molecular caryotype performed? I understand that in the year 2000 was not an option, but in these past five years is mandatory in a CHD?
ANSWER: Thank you. As you already mentioned in early years it was not possible. Over the last years we performed it. As requested by you, we clarified this now in the revised Methods section.
Line 218: “in patients with dexamethasone” Redundant.
ANSWER: Thank you for this important comment. We removed the phrase and changed the sentence.
Patients who received dexamethasone hat a significantly lower gestational age at birth (Pearson Correlation -0.361, p= 0.008) with a median gestational age of 34.7 weeks (32-35.3) compared to the gestational age of 38 weeks (35-39.4) in patients without dexamethasone treatment.
Line 311: [4][12][29][30][30]. Please check the references
ANSWER: Thank you. We double checked and changed the references accordingly.
Line 323: [31][10][26]. Please, put them in the proper order.
ANSWER: Thank you. We put the references in the proper order.
Line 332: [33][8][9][31]. Please, put them in the proper order.
ANSWER: Thank you. We put the references in the proper order.
Line 356: The surname is wrong: Escobar
ANSWER: Thank you. We are sorry for this mistake. We changed it.
A conclusion section should be introduced.
ANSWER: Thank you for this valuable suggestion. We have now added a conclusion section in the revised version of the manuscript.
Reviewer 3 Report
Thsi is a single center retrospective cohort study assessing the rate of stillbirth and early and late neonatal mortalty for pregnancies complicated by fetal dysrhythmias across two decades. The research question is interesting, the methodology appropaiate and results are overall sound and interesting. However, I have constructive criticisms for the authors, before publication.
1. In this series there is a significant risk of preterm birth. I suppose that this is a istrogenic preterm birth done by elective or urgent cesarean section due to fetal haemodynamic deterioration. Were there spontaneous preterm delivery? Please add a comment and detail on this issue.
2. In the multivariate model cardiac decompensation does not correlate with the outcome neonatal death. This is unusual and I believe it may depend on the timing of assessment (as decompensation may occur later on) as well as to its definition (do all cases have the same severity?). The definition was provided, however dicotomizing such a complex outcome may have determined failure to defining its predictive capability. Is it possible to create a continuous variable for this outcome "cardiac decompensation" in different classes of severity? Is getational ge at assessment homogeneous in the all group?
3. Please briefly present and discuss the important role of appropriateness of treatment in order to depict differences in neonatal outcome. A recent important reference on treatment may be considered (cit n 1)
4. It is difficult to imagine a fetus with irregular rythm due to extrasystole, undergoing TOP or neonatal death without additional explanations. Please clarify well in the conclusions that TOP and neonatal death were due to additional problems of these cases (CHD mainly) and please present the outcome in the groups with isolated arrithmia vs arrithmia associated to additional defects.
5. Another interesting question wuold be to discuss/assess the risk of neonatal death of specific fetal abnormalities (CHD specifically) with or withour associated arrhytmia. Can they speculate on that? Can effective treatment reduce the risk? These may be open questions for future research.
References
1. Veduta A, et al. Treatment of Fetal Arrhythmias. J Clin Med. 2021 Jun 6;10(11):2510. doi: 10.3390/jcm10112510. PMID: 34204066; PMCID: PMC8201238.
Author Response
- In this series there is a significant risk of preterm birth. I suppose that this is a istrogenic preterm birth done by elective or urgent cesarean section due to fetal haemodynamic deterioration. Were there spontaneous preterm delivery? Please add a comment and detail on this issue.
ANSWER: Thank you very much for this valuable comment. We have now included the following sentence in the revised version of the Results section.
A total of 38 children were born preterm before 37 weeks of gestation, mainly iatrogenic due to fetal deterioration.
- In the multivariate model cardiac decompensation does not correlate with the outcome neonatal death. This is unusual and I believe it may depend on the timing of assessment (as decompensation may occur later on) as well as to its definition (do all cases have the same severity?). The definition was provided, however dicotomizing such a complex outcome may have determined failure to defining its predictive capability. Is it possible to create a continuous variable for this outcome "cardiac decompensation" in different classes of severity? Is gestational age at assessment homogeneous in the all group?
ANSWER: Thank you for your comment. We absolutely agree with you that this lack of correlation between cardiac decompensation and neonatal death was unusual. In the univariate analysis, neonatal death correlated with fetal decompensation. This correlation was not found in the multivariate analysis. This may be caused by relatively low patient numbers in each subgroup. Additionally, due to local guidelines, patients were monitored at short intervals. If signs of fetal decompensation were observed and if fetal intrauterine therapy showed no effect, prematurity was more likely to be accepted than fetal demise.
As you state, cases with cardiac decompensation did not have the same level of severity. Some fetuses showed pericardial effusion, some ascites, some cardiomegaly and some hydrops fetalis. It was also possible that one fetus had several signs. To the best of our knowledge an official classification, which can be converted in a continuous variable, does not exist.
Since dysrhythmias occurred in different weeks of gestation, it was not possible to perform a homogenous assessment.
- Please briefly present and discuss the important role of appropriateness of treatment in order to depict differences in neonatal outcome. A recent important reference on treatment may be considered (cit n 1)
ANSWER: Thank you for this thoughtful comment. According to your comment, we have now included the following statements in the revised version of the manuscript (Discussion section):
The decision for intrauterine fetal treatment depends on several factors, like the type and severity of dysrhythmia, gestational age, associated CHD and fetal signs of decompensation. Choosing the right therapy can have a major impact on children's outcome. In general, the motivation and range for treatment in utero increases with prematurity (32). In the beginnings of transplacental drug application, digoxin has been established as first-line therapy in the treatment of fetal tachyarrhythmias. For successful digitization, the dose is saturated to a digoxin level of 2 – 2.5 ng/ml either intravenously or orally, then a maintenance dose of 0.25 mg every 8 hours should be applied orally (22)(33). However, digoxin should not be given in cases where fetal hydrops has been diagnosed as the distribution in decompensate fetuses is reduced (4). Over the years other medications showed effective treatment of fetal tachycardia. Due to higher rates of adverse side effects some medications are not recommended for standard use. Verapamil was described with presence of sudden demise, whereas procainide may trigger preterm delivery. Currently, besides digoxin, flecanoids and soltalol are used as standard therapy (34). Flecainide 100mg every 8 hours can lead to conversion of fetal supraventricular tachycardia (8)(9)[31](34). Miyoshi et al. established a protocol for pregnancies without signs of hydrops (12). Jaeggi et al initiated the FAST trail, where patients with different types of tachyarrhythmia are randomized and receive prespecified therapy protocols (35). Outcomes will be examined in a prospective analysis (35).
Later in the Discussion section we added following sentence:
The American Heart Association recommends dexamethasone treatment only in fetuses with severe AV-block, while other forms of bradyarrhythmia should be closely observed (40).
Current references have been added.
- It is difficult to imagine a fetus with irregular rythm due to extrasystole, undergoing TOP or neonatal death without additional explanations. Please clarify well in the conclusions that TOP and neonatal death were due to additional problems of these cases (CHD mainly) and please present the outcome in the groups with isolated arrithmia vs arrithmia associated to additional defects.
ANSWER: Thank you for your comment. We described in our Discussion section that TOP and neonatal death in patients with rhythm irregularities occurred only in patients with additional CHD. Furthermore, the supplementary table shows the characteristics of neonates with neonatal death. In the conclusion, we comment that the outcome of rhythm irregularities is generally by far more favorable, especially when no CHD is detected.
5. Another interesting question wuold be to discuss/assess the risk of neonatal death of specific fetal abnormalities (CHD specifically) with or withour associated arrhytmia. Can they speculate on that? Can effective treatment reduce the risk? These may be open questions for future research.
ANSWER: Thank you for your comment. We added following comment in the Discussion section:
As we know, structural heart disease can lead to fetal arrhythmia, especially to ex-citation conduction disorders, such as AV-block. This work highlights the role of CHD as an unfavorable factor for survival in fetal arrhythmias. Further research with larger case numbers per specific CHD is needed to determine whether the development of arrhythmias in fetuses with CHD leads to higher rates of adverse offspring outcomes, compared to fetuses with the same specific CHD but without intrauterine or neonatal arrhythmia.
References
- Veduta A, et al. Treatment of Fetal Arrhythmias. J Clin Med. 2021 Jun 6;10(11):2510. doi: 10.3390/jcm10112510. PMID: 34204066; PMCID: PMC8201238.
Reviewer 4 Report
Dear authors, it has been a big pleasure for me to review this paper
the topic is absolutely interesting, the study well conducted and the paper well written
I have just some minor revision to suggest
1) pay attention to all references as an example [4][12][29][30][30] line 311
2) would suggest to add a flow chart to describe your usual management for fetal tachicardia and bradicardia
best regards
Author Response
ANSWER: Thank you very much for acknowledging our work. We are very pleased about your comment.
- pay attention to all references as an example [4][12][29][30][30] line 311
ANSWER: Thank you. We checked and changed the references.
2) would suggest to add a flow chart to describe your usual management for fetal tachicardia and bradicardia
ANSWER: Thank you for this valuable comment. After discussing this point with all authors, however, we felt that the paper might get too lengthy by adding a flow chart – there are already six tables included in the main body of the manuscript.
We do strongly agree with you that clarification of our usual management in fetal tachycardia and bradycardia should be included. Therefore, we have now added the following statements in the Methods section of the revised version of the manuscript (lines 123 – 127):
´In short, fetuses with bradycardia and maternal Ro antibodies were primarily treated with dexamethasone, while the first line treatment for fetal tachycardia consisted of digoxin, followed by flecainide in some cases. Fetuses with other rhythm irregularities were primarily followed up in weekly or bi-weekly intervals.´
If you do feel, however, that the insertion of a flow chart on the usual treatment modalities at our center would tremendously increase the quality of the paper, please be so kind and let us know as soon as possible and we will provide it instantly.
Reviewer 5 Report
Minor language corrections are needed. More recent references would be helpful
Author Response
Thank you for your suggestions. The article was now reviewed by a native speaker, language corrections were performed, and more recent references have been added in the revised version of the manuscript.